# Overview of Modern Commercial Kits for Laboratory Diagnosis of African Swine Fever and Swine Influenza A Viruses

**DOI:** 10.3390/v16040505

**Published:** 2024-03-26

**Authors:** Larysa Muzykina, Lucía Barrado-Gil, Antonio Gonzalez-Bulnes, Daniel Crespo-Piazuelo, Jose Joaquin Cerón, Covadonga Alonso, María Montoya

**Affiliations:** 1Molecular Biomedicine Department, The Margarita Salas Centre for Biological Research (CIB) of the Spanish National Research Council (CSIC), C. Ramiro de Maeztu, 9, 28040 Madrid, Spain; larysa.muzykina@cib.csic.es; 2Department of Biotechnology, INIA-CSIC, Centro Nacional Instituto Nacional de Investigación y Tecnología Agraria y Alimentaria, Ctra. de la Coruña Km 7.5, 28040 Madrid, Spain; barrado.lucia@inia.csic.es (L.B.-G.); covadonga.alonso@csic.es (C.A.); 3R&D Department, Cuarte S.L., Grupo Jorge, Ctra. de Logroño km 9.2, Monzalbarba, 50120 Zaragoza, Spain; antoniogbulnes@cuartesa.com (A.G.-B.); danielcrespo@cuartesa.com (D.C.-P.); 4Interdisciplinary Laboratory of Clinical Analysis (Interlab-UMU), University of Murcia, 30100 Murcia, Spain; jjceron@um.es

**Keywords:** swine influenza A viruses, African swine fever virus, diagnostic tests, lateral flow devices, ELISA test, real-time PCR, RT-PCR

## Abstract

Rapid and early detection of infectious diseases in pigs is important, especially for the implementation of control measures in suspected cases of African swine fever (ASF), as an effective and safe vaccine is not yet available in most of the affected countries. Additionally, analysis for swine influenza is of significance due to its high morbidity rate (up to 100%) despite a lower mortality rate compared to ASF. The wide distribution of swine influenza A virus (SwIAV) across various countries, the emergence of constantly new recombinant strains, and the danger of human infection underscore the need for rapid and accurate diagnosis. Several diagnostic approaches and commercial methods should be applied depending on the scenario, type of sample and the objective of the studies being implemented. At the early diagnosis of an outbreak, virus genome detection using a variety of PCR assays proves to be the most sensitive and specific technique. As the disease evolves, serology gains diagnostic value, as specific antibodies appear later in the course of the disease (after 7–10 days post-infection (DPI) for ASF and between 10–21 DPI for SwIAV). The ongoing development of commercial kits with enhanced sensitivity and specificity is evident. This review aims to analyse recent advances and current commercial kits utilised for the diagnosis of ASF and SwIAV.

## 1. Introduction

Nowadays, the global demand for meat is surging, and pork stands as a major economic force worldwide. Throughout history, outbreaks of transmissible diseases have left a significant impact on human health and livelihood. Recent viral outbreaks affecting both humans and animals have brought attention to the focus on the persistent threat of viral epidemics. Examples of this situation are the episodes of African swine fever (ASF), causing detrimental effects on swine production, and the dissemination of influenza A viruses (IAV) in both pigs and human populations.

Given the current scenario of emerging pathogens, the early detection of viruses affecting pigs is imperative for the timely implementation of control measures that mitigate the spread of diseases. An effective strategy in the battle against infectious diseases lies in the constant monitoring of their spread through the use of point-of-care (PoC) tests and laboratory methods. These methods encompass basic rapid test kits for detecting antigens or antibodies employing lateral flow devices, enzyme-linked immunosorbent assays (ELISAs) or highly sensitive viral DNA/RNA detection techniques.

ASF is one of the most devastating diseases affecting domestic pigs and wild boars, leading to a considerable economic burden [1]. Swine influenza is a highly significant respiratory disease in swine. The primary causative agents are swine influenza A viruses (SwIAV), specifically the H1N1, H1N2, and H3N2 subtypes, which are antigenically related to human IAV [2]. Therefore, the spread of swine viral diseases affecting the respiratory system, and their potential impact on humans, also requires special control measures and joint surveillance [3,4].

The purpose of this review is to provide insights into the current importance of ASF and SwIAV, and the updates on diagnostic improvements. This includes the information and discussion on new commercial kits for the detection of specific antibodies and antigens, alongside viral genome analysis through quantitative real-time PCR or isothermal amplification assays targeting the causative agents of these swine diseases.

## 2. African Swine Fever (ASF)

Currently, ASF is considered a major emerging threat, given its increased incidence and spread from Central and Eastern Europe to Southeast Asia, America and Oceania, causing a high economic burden and impacting animal health [5,6,7]. In domestic pigs and wild boars, ASF usually results in severe infectious disease with a case fatality rate close to 100%. It is characterised by fever, haemorrhagic diathesis of the skin and internal organs, as well as dystrophic and necrotic changes in parenchymal organs. The etiological agent causing ASF is the African swine fever virus (ASFV), the sole member of the *Asfarviridae* family and is part of the nucleocytoplasmic large DNA viruses (NCLDV) group [8].

The disease caused by this virus can manifest in various forms, including peracute, acute, subacute or chronic. The acute form of the disease courses as an haemorrhagic fever. Affected animals exhibit symptoms such as fever, a tendency to crowd, loss of appetite, inactivity and apathy. Early laboratory analysis reveals signs of leukopenia induced by lymphopenia, along with alterations in monocyte and immune cell numbers [9,10,11]. However, less virulent isolates that emerged during the circulation of the virus in domestic pigs have led to an increase in subacute and unapparent infections, thereby increasing the prevalence [12,13,14,15]. Virulent isolates responsible for the acute form may induce 100% mortality within 5–15 days. Moderately virulent isolates give rise to the subacute form, characterised by reduced mortality and recovery in up to 50% of animals. Low virulence isolates can lead to chronic disease, displaying characteristic clinical signs such as arthritis, skin lesions, enlarged lymph nodes, and renal dysfunction. Attenuated and non-pathogenic isolates may cause few disease signs or no disease. Moreover, the virus can persist for prolonged periods in tissues or blood from recovered pigs. Following infection with low virulence isolates, this persistence might contribute to virus transmission, disease persistence, sporadic outbreaks and ASFV introduction into disease-free zones [16,17].

ASFV enters the body via the tonsils or dorsal pharyngeal mucosa and then migrates to the mandibular or retropharyngeal lymph nodes; from here, the virus spreads systemically. Subsequently, the virus becomes detectable in nearly all pig tissues [18].

ASFV exhibits a complex icosahedral multilayer structure, comprising the envelope, capsid, inner envelope, core–shell, and nucleolus [19]. It encodes 68 structural proteins and over 100 non-structural proteins [20]. The investigation of structural characteristics, functions, and molecular mechanisms of viral proteins serves as a foundation for the development of diagnostic kits and vaccines. Currently, 24 genotypes are identified through partial sequence analysis of the C-terminal region of the *B646L* gene, which encodes the p72 viral capsid protein [21]. Due to the stability of the p72 gene in some field isolates from different outbreaks, a recent study classifies ASFV into six genotypes based on p72 amino acid sequences [22].

Although scientists have been familiar with this disease for over a century only two safe and effective ASF vaccines have received approval from a regulatory body. These two vaccine candidates, ASFV-G-ΔI177L [23,24] and ASFV-G-ΔMGF [25,26,27] are based on ASF-modified live virus (MLV) and have been licensed for field use in Vietnam [28]. Inactivated vaccines are not available due to the difficulty of inducing immunity while meeting minimum efficacy standards. Viral vector-based vaccines, subunit vaccine candidates and DNA vaccines that can be produced in scalable vaccine platform expression are being evaluated in ongoing laboratory research in experimental challenge models. Although vaccination would be the optimal strategy, most of the approaches are still in the development phase [29,30]. For this reason, an effective approach to combat ASF involves early diagnosis followed by the elimination of the causative agent. The damage caused by the disease extends beyond the loss of pig herds and their products to include significant restrictions in international trade. Although ASF primarily affects only pigs and is relatively safe for humans, the appearance of ASF in any country can be considered a socioeconomic impact on the state [31].

### 2.1. Diagnosis

The prompt control of African swine fever is highly dependent on accurate diagnosis, especially considering the increasing prevalence and continual emergence of variant strains. Haemadsorption test (HAD), conventional PCR and real-time PCR, antigen detection by fluorescent antibody test and enzyme-linked immunosorbent assay (ELISA) are among the most widely used methods for detecting ASFV.

#### 2.1.1. Virus Detection

HAD, which is conducted in primary leukocyte cultures, proves effective in detecting ASFV [32]. However, the fact of taking several days to obtain results and collection of primary cells is a laborious process. Indeed, a critical step for this method is the condition of the sample under study. In addition, attenuated and non-haemadsorbing (non-HAD) isolates may give false negative results [33].

#### 2.1.2. Genome Detection

Conventional PCR is recognised by the WOAH (World Organisation for Animal Health) as a specific and rapid method for the detection of several ASFV strains throughout the structural protein VP73 [34].

Real-time PCR is considered the standard method for the early diagnosis of ASF due to its superior sensitivity, specificity, and high throughput capability for the detection of the ASFV genome from clinical samples obtained from domestic pigs, wild boars, and ticks [35].

In Belgium, the sensitivity and specificity of seven commercially available ASFV real-time PCR kits and three commercially available Taq polymerase reaction mixes were evaluated by Schoder et al. [36]. Their results are presented in Table 1.

The precision, specificity, sensitivity, and accuracy of the various kits were greater than 95% for all methods, making them well-suited for the detection of genotype II ASFV isolates.

At the German National Reference Laboratory (NRL), eight commercial real-time PCR systems underwent testing using the same set of experimental samples. The suitability for use was established for the following PCR tests: INgene qPPA (Gold Standard Diagnostics), ASFV virus type (Indical Bioscience, former Qiagen Leipzig), ID Gene ASF Duplex (IDvet), RealPCR ASFV (IDEXX), SwineFever combi (Gerbion), ViroReal kit ASFV virus (Ingenetix), Kylt ASF Real-time PCR (Anicon) and VetMAX African Swine Fever Virus detection kit (Thermo Fisher Scientific). According to the analysis of the results, all kits were suitable and provided reliable results [37].

In accordance with the WOAH procedure for the registration of diagnostic kits and following the recommendations of the WOAH Biological Standards Commission, VetMAX™ African Swine Fever Virus Detection kit (Thermo Fisher Scientific) is certified for ASF virus detection from blood, serum and tissues from both domestic and wild swine. Recent results indicate that this kit has a diagnostic sensitivity (DSe) of 100% (51 tissues) and a diagnostic specificity (DSp) of 100% (1563 blood, serum and 63 tissues) [38].

African swine fever and classical swine fever (CSF) have similar clinical manifestations and pathological changes, making the laboratory differential diagnosis of these two diseases highly relevant in the field. The first development of a one-step triplex real-time PCR assay for the simultaneous differential detection of two viruses appeared in 2013. In this study, the authors combined a modified version of the classical swine fever virus (CSFV) detection protocol, incorporating an exogenous internal control, with a new real-time assay for ASFV. As a result of this research, the diagnostic sensitivity was 100% for both viruses, while the diagnostic specificity was estimated to be 100% for CSFV detection and 97.3% for ASFV detection [39].

Commercial kits for ASF diagnosis, produced by well-known global manufacturers, can be too expensive for some laboratories. Therefore, scientists from several countries are actively involved in developing kits at more affordable prices, prioritizing effectiveness while maintaining high diagnostic accuracy. One such example is the “ASF/CSF duo qRT-PCR” kit, designed for the differential diagnosis of ASF and CSF in Ukraine. This kit was validated at the Loeffler Institute (Rims Island, Germany) and achieved high-sensitivity yields (five copies/reaction). The diagnostic sensitivity and specificity for ASFV detection were 97.5% and 100%, respectively. Similarly, for CSFV detection, these features reached 100% in both cases [40].

Recently, the ASFV MONODOSE dtec-qPCR kit (GPS™, Orihuela, Spain) was validated and the results were compared with two reference methods based on qPCR [41,42], as recommended by the WOAH. The diagnostic validation involved the analysis of 181 pig samples obtained from whole blood, serum, kidney, heart, liver and tonsils. The ASFV MONODOSE dtec-qPCR kit demonstrated 100% diagnostic specificity and sensitivity for all samples, aligning with the findings of Fernández-Pinero et al. [43].

The main disadvantage of these techniques was the dependence on expensive thermal cyclers. Consequently, isothermal amplification assays, including loop-mediated isothermal amplification (LAMP) and recombinase polymerase amplification (RPA) assay, have been developed for detecting ASFV. The clustered regularly interspaced short palindromic repeats (CRISPR)-associated endonuclease Cas, derived from prokaryotic immune systems, serves as a versatile tool with several applications. Endonucleases like Cas12a/b, Cas13a/b and Cas14 possess random cleavage of single-stranded DNA or RNA [44]. Due to the spread of ASFV in Asia, scientists from Sun Yat-sen University and Guangzhou University have developed a highly sensitive new assay based on CRISPR-Cas12a for ASFV detection. To identify different ASFV genotypes, they designed 19 crRNAs targeting the conserved p72 gene of ASFV and found several high-activity crRNAs for potential use in case of the emergence of new ASFV variants. Research data indicates the CRISPR-Cas12a-based assay exhibits a sensitivity approximately ten times higher than that of a commercial quantitative PCR (qPCR) kit or the WOAH-recommended qPCR. The CRISPR-Cas12a-based assay demonstrated specific detection of ASFV without cross-reactivity with other critical swine viruses and different virus genotypes. It was observed that an extended incubation time corresponds to a higher detection limit, making it applicable for detecting weakly positive samples and new ASF variants. In addition, both the CRISPR-Cas12a-based assay and the commercial qPCR displayed excellent agreement [45].

#### 2.1.3. Serological Diagnosis

Serological diagnosis of ASF is critical for identifying asymptomatically infected animals and those that have recovered, as antibodies to ASF typically appear 7–10 days post-infection and can persist for months or even years. The presence of specific antibodies indicated infection. Naturally occurring attenuated ASFV strains have been identified in recent years, causing non-lethal or subacute disease, which is challenging to identify by molecular-based diagnosis methods.

In scientific institutions and leading diagnostic laboratories, ELISA along with alternative tests such as indirect fluorescent antibody technique (IFAT), immunoperoxidase technique (IPT) or immunoblot (IB) are the most widely used methods for detecting antibodies against ASF. In most diagnostic veterinary laboratories, commercial tests are employed to detect the presence of antibodies or antigens to ASF [46].

The results of a study using a commercial ELISA for virus detection as INgezim PPA DAS K2, (Gold Standard Diagnostics, Madrid, Spain) showed good to moderate agreement with WOAH-recommended PCR methods (κ = 0.67 [95% CI, 0.58 to 0.76]) with a sensitivity of 77.2% [47].

In 2014, the European Union Reference Laboratory (EURL) for ASF conducted a study to compare the WOAH indirect ELISA based on the ASFV semi-purified antigen and three commercial ELISA diagnostic kits [47]. The specificity of these kits is presented in Table 2.

The confirmatory immunoperoxidase test (IPT) has proven to be more sensitive than ELISA, as it allows the detection of antibodies to ASFV at an earlier stage of the infection. The WOAH recommends all positive ELISA results should always be confirmed by alternative methods, such as IPT or immunoblotting (IB) [47].

#### 2.1.4. Rapid Test Kits

There are validated commercial kits available for field testing, which include basic antigen or antibody rapid test kits using lateral flow devices. These are easy to use and can provide results within 15–30 min. These rapid tests are used to detect antibodies or antigens, but they are usually less sensitive than ELISA and PCR tests. Simple rapid tests can be used to establish a preliminary diagnosis when ASF is suspected in production conditions (farm or pork processing plant).

The WOAH ASF Reference Laboratory Network for African Swine Fever diagnosis performed a study using four point-of-care (PoC) test methods for rapid detection of ASFV antigen and three for rapid detection of antibodies [48]. A summary of information on diagnostic sensitivity (DSe) and specificity (DSp) together with each original reference is presented in Table 3.

The INgezim ASF CROM Ag test results, employing the lateral flow assay (LFA) for p72 antigen detection, demonstrated an excellent correlation between LFA and ELISA. However, PCR consistently exhibited higher sensitivity (38% positive samples by PCR vs. 27% by LFA). In comparison to antigen-ELISA, this kit yielded 60% of positive results, whereas ELISA had 48%. The diagnostic sensitivity of this test was 67.86%, and the diagnostic specificity was 97.98% [49,50].

German scientists have observed a sensitivity decrease to only 12.5% when employing the lateral flow analysis of INgezim ASFV Crom Ag with blood samples of wild boars in conditions resembling field settings. These results highlight the necessity for enhancing LFA antigen for testing wild boar carcasses at the point of detection [56].

According to the manufacturer’s instructions, the Rapid ASFV Ag Test kit can detect structural antigens (proteins p72, p32) of the ASFV virus in serum, plasma or whole blood of pigs. Available data on their web page [52] demonstrate, that the Bionote ASFV Ag Test kit exhibits “outstanding sensitivity and excellent specificity (100% Specificity when tested in a ASF non-outbreak region)”. This kit underwent testing by The Australian Centre for Disease Preparedness (ACDP) [48] and was compared to in-house ASF PCR described by Zsak et al. [57]. The results indicated that “68% of PCR positive samples were positive; 90% of PCR negative samples were negative”.

PenCheck’s Antibody-Based Lateral Flow Assay (LFA) effectively detects the majority of ASF-positive pigs during the initial days of infection. The samples used in the PenCheck LFA studies included blood, serum, and tissue samples collected from pigs experimentally infected with ASFV, as well as whole blood, plasma and tissue samples from the field. The limit of detection of the assay was an ASFV titer 10^7.80^ TCID _50_/mL, corresponding to ASFV real-time PCR cycle threshold (Ct) below 23. Due to low viral loads, sample types such as nasal, oral and rectal swabs are not suitable for testing using the PenCheck LFA^®^. Consequently, this kit can effectively detect infected pigs with virulent ASFV strains that have characteristic clinical signs but is unlikely to detect pigs infected with low-virulence ASFV strains [53].

An ASF antigen detection kit from Shenzhen Lvshiyuan Biotechnology Co. Ltd. (SLB, Shenzhen, China) underwent evaluation using clinical field samples submitted to the National Animal Health Laboratory (NAHL) from ASF suspect cases obtained in Laos in 2019. Positive and negative samples of whole blood and homolysed serum were assessed by rapid diagnostic test (RDT) and PCR, with the latter serving as the gold standard reference comparator. The overall SLB ASF RDT accuracy of results, when combining all sample types, was DSe 65%, 95% CI (51–77) and DSp 76%, 95% CI (62–87) with the positive predictive value (PPV) 76%, and the negative predictive value (NPV) 66% [54].

The Herdscreen^®^ ASFV Ab Test uses p30, an early ASFV protein. This antigen is produced at the company Algenex (Madrid, Spain) using CrisBio^®^ technology, which employs insects as living biofactories in combination with baculovirus vectors. According to information from the Algenex website, in 2020 a validation study of this kit was conducted at the European Union Reference Laboratory for African Swine Fever (EURL-ASS). In a validation study using 282 serum and blood samples from ASFV-positive animals, the Herdscreen^®^ ASFV Ab Test demonstrated an analytical specificity of 100% and an analytical sensitivity of 86%. Thus, it is recommended for the initial diagnosis of ASF in the field for both serum and blood samples [55]. However, these data are not peer-reviewed. Obtaining a positive result with ASF rapid tests requires confirmation by laboratory diagnostic tests described in Chapter 3.9.1 of the WOAH Terrestrial Manual [46].

## 3. Influenza A Viruses of Swine (SwIAV)

Swine influenza A (SwIA) is a highly contagious viral infection of pigs. The disease typically spreads rapidly within swine units, even though all infected pigs may show clinical signs, followed by a swift recovery of the affected animals. SwIA generally does not pose a significant challenge to pig production unless accompanied by other complications. However, strict surveillance is essential, considering its potential transmission to humans or other animals. The causative agents of swine influenza are influenza A viruses, with the most common subtypes being H1N1, H1N2 and H3N2 [58]. The evolution of SwIAV involves adaptation to different hosts, antigenic drift and genetic reassortment, exhibiting both enzootic and geographic dependence. Detailed mechanisms of influenza virus evolution in Europe, Asia and America are expounded in scientific studies and summarised in the following reviews [59,60,61,62,63,64,65].

The transmission of infection occurs through contact with sick animals, particularly with secretions containing viral particles in aerosols generated by coughing, sneezing and nasal discharge. SwIAV infection can, in some cases, lead to reproductive problems and abortions. While mortality is typically low, morbidity reaches 100%, and the severity of SwIAV can be exacerbated by secondary bacterial infection.

Detailed information regarding the in vitro interaction between influenza viruses and the porcine immune system, along with in vivo immune responses during influenza infection in pigs with different subtypes was described in previous reviews [2,66,67].

Lewis, Russell et al. conducted an analysis of the antigenic variation in hundreds of H1 and H3 viruses found in pigs across multiple continents. A significant factor contributing to the diversity of the H1 and H3 viruses in pigs is the frequent introduction of human viruses. In contrast, only one influenza A virus originating from a bird substantially contributed to the observed antigenic diversity in pigs in. Once introduced to pigs, influenza A viruses derived from humans undergo continual antigenic mutations, resulting in a remarkable diversity of viruses. These viruses can be transmitted not only among pigs but also to humans, posing a serious risk to public health due to their divergence from current human influenza A virus strains [68,69]. 

### 3.1. Diagnosis

Haemagglutination inhibition (HI) and neuraminidase inhibition (NA) tests, reverse transcription–polymerase chain reaction (RT-PCR), and ELISA, are among the most widely used methods for detecting SwIAV.

#### 3.1.1. Virus Detection

Tracheal, lung and nasal swabs are essential to detecting and identifying pathogens and should be collected within 24–72 h of the development of clinical signs. Alternatively, oral fluids collected from cotton ropes suspended in the pigsty can also be useful as a group or population sample. For virus isolation, cultivation of embryonic chicken eggs and cell lines is employed. The classical methods for typing influenza viruses (HxNy) are the identification of haemagglutinin (HA) and neuraminidase (NA) involve identifying haemagglutination inhibition and neuraminidase inhibition tests performed on isolated viruses [70].

#### 3.1.2. Reverse Transcription–Polymerase Chain Reaction (RT-PCR)

The fastest and most accurate method for detecting viral RNA and subtype SwIAV virus in veterinary diagnostic laboratories is through RT-PCR analysis. This technique targets matrix (M), haemagglutinin (HA), neuraminidase (NA) and nucleoprotein (NP). Additionally, commercially available test kits are widely utilised.

Recent developments and enhancements in molecular diagnostic methods have established them as the primary choice for diagnosing SwIAV infections. An advantageous aspect of using readily available, commercially produced, highly sensitive kits for detecting viral RNA and confirming the laboratory diagnosis of SwIAV is the presence of internal test standards and positive controls for the relevant serotypes. However, caution is warranted due to the heightened sensitivity of RT-PCR, which can detect viral RNA in samples, even in the absence of an infectious virus. Therefore, interpreting results with low detection limits requires careful consideration, as they may not necessarily indicate an active infection. VetMAX™-Gold SIV Detection assay is a single-tube (or well-plate) real-time RT-PCR assay in which swine influenza virus and Xeno™ internal positive control RNA are reverse transcribed into cDNA, then amplified and detected in real-time using fluorescent TaqMan™ probes. The VetMAX™-Gold SIV Subtyping kit is a multiplex, one-step qRT-PCR kit for the detection and differentiation of the predominant subtype strains of SwIAV. This enables early detection in pigs, contributing to minimising the spread of the disease caused by the virus. Consequently, both USDA-licensed kits underwent a study establishing their sensitivity and specificity [71]. The results are presented in Table 4.

Most scientific and diagnostic veterinary laboratories have recently adopted WOAH-recommended specific primers and probes for the swift detection and identification of SwIAV subtypes using real-time RT-PCR. It is a specific, sensitive and cost-effective way to obtain rapid diagnosis. The specific primers encode the matrix (M) or nucleoprotein (NP), which are highly conserved across all IAV viruses. These components serve as optimal targets for SwIAV screening infection via RT-PCR, followed by specific real-time testing for H1 and H3 subtype viruses.

Dr. Marek J. Slomka et al., as early as 2010, described molecular assays based on real-time reverse transcription–polymerase chain reaction (RRT-PCR) of the avian influenza M gene, which was adapted for use in pigs. The “perfect match” M gene RRT-PCR emerged as the most sensitive variant of this test for the detection of established European SwIAVs and H1N1v. H1 RRT-PCR specifically identified H1N1v but not European SwIAVs. This “perfect match” M gene RRT-PCR exhibited 100% sensitivity and 95.2% specificity for swabs, 93.6% sensitivity and 98.6% specificity for tissues. The H1 RRT-PCR demonstrated sensitivity and specificity of 100% and 99.1%, respectively, for swabs, and 100% sensitivity and 100% specificity for tissues [72].

Furthermore, scientific studies detailing the development of specific primers and probes based on HA and NA for the differential diagnosis of SwIAV subtypes are already available [73,74,75].

#### 3.1.3. Serological Diagnosis

The “golden standard” serological test for detecting SwIAV antibodies is the haemagglutination inhibition (HI) test, conducted on paired sera and specifically targeting the HA subtype. Ideally, sera are collected 10–21 days apart. A four-fold or greater titer increase between the first and second samples suggests a recent SwIAV infection. Other described serological tests include agar gel immunodiffusion test, indirect fluorescent antibody test, virus neutralisation, and ELISA. Due to the growing antigenic diversity in SwIAV and the necessity to incorporate multiple H subtypes in HI assays, there is a general shift towards using commercially available ELISAs that are not subtype-specific [69].

In 2016, a study on screening assays for detecting serum antibodies to SwIAV revealed that the HI assay for antibody detection may not always be useful “because of the biological vagaries of the assay”. The commercial indirect ELISAs for H1N1 and H3N2 (IDEXX SIV H1N1 Ab Test, IDEXX Laboratories, Inc., Westbrook, ME, USA; IDEXX SIV H3N2 Ab Test; IDEXX Laboratories, Inc.) utilised in this study exhibited low diagnostic sensitivity (1.4% and 4.9%, respectively), and are not recommended for the detecting antibodies against current strains of SwIAV. However, when employing a commercially available ELISA that blocks IAV nucleoproteins (NPs) (IDEXX AI MultiS-Screen Ab Test; IDEXX Laboratories, Inc.) with a threshold S/N ≤ 0.60, the sensitivity and specificity were 95.5% and 99.6%, respectively [76].

Yoshikazu Fujimoto et al. conducted a serological survey of influenza A virus infection in Japanese wild boar. They determined the subtype-specific seroprevalence of anti-influenza antibodies in collected serum samples using a serum virus neutralization (SVN) assay and a commercially available «Influenza A Ab test» ELISA kit (IDEXX). Researchers observed that if using a cut-off factor of 0.50, following the manufacturer’s recommendations, only 1 sample out of 59 tested positive. However, using Youden’s index, the optimal threshold S/N ratio for Japanese wild boar serum in this study was determined to be 0.81, resulting in 12 positive samples for IAV infections out of 59 samples tested. The authors highlighted the importance of setting a high threshold for the S/N ratio to efficiently detect positive wild boar serum against IAV infection when utilising commercial ELISA kits [77].

According to research by the manufacturer Gold Standard Diagnostics, 24 samples of animals vaccinated with H1N1, H1N2, H3N2, or a combination of these three strains were analysed during the development of the INgezim Influenza Porcina kit (R.11.FLU.K1). The results from the study demonstrated 87% sensitivity and 89% specificity compared to IHA [78]. This kit was then utilised to investigate the infectious disease reservoir in Greek European wild boar during the 2006–2010 hunting seasons in different regions of Greece [79]. 

The ID Screen^®^ Influenza A Nucleoprotein Swine Indirect ELISA is designed to detect antibodies directed against the nucleoprotein of IAV in swine serum or plasma. This commercial quantitative indirect ELISA proves effective in detecting natural infection and monitoring killed vaccines in swine. It offers improved sensitivity compared to competitive ELISA and HI tests. The analysis was conducted at the National Reference Laboratory for Swine Influenza in France and at IDvet. A measured sensitivity of 100% (CI95%: 98.76–100%) was established based on 400 swine sera from field animals. Additionally, 15 sera samples from vaccinated or infected swine were tested, resulting in a sensitivity of 100% (CI95%: 75.75–100%) for the ID Screen^®^ Influenza A Nucleoprotein Swine Indirect, and 86.67% (CI95%: 62.12–96.27%) for the ID Screen^®^ Influenza A Antibody Competition Multi-species. The specificity of the assay was assessed using 343 experimental and field sera from non-vaccinated pigs. All sera were classified as negative, yielding a measured specificity of 100% (CI95%: 98.89–100%) [80].

In addition to detecting post-infectious antibodies, ELISA test kits are employed for the identification of maternally derived antibodies (MDAs) and post-vaccination antibodies. Vaccination against SwIAV is primarily administered in breeding herds for sows and for immunising growing pigs at weaning. In one of the most recent studies, the PrioCHECK™ Swine Influenza Ab Serum Plate kit (ThermoFisher Scientific, Lelystad, the Netherlands) was utilised to detect antibodies against SwIAV in piglets with different MDA statuses following a single-dose vaccination [81].

#### 3.1.4. Rapid Test Kits

Rapid test kits designed for preliminary screening of SwIA viral antigen have been developed, but their number, compared to rapid tests for diagnosis of influenza A in humans and birds, is much smaller. Chapter 3.9.7 of the WOAH contains information about the possibility of using type A antigen-capture ELISAs and membrane immunoassays for the detection of SwIAV in lung tissue and nasal swabs. However, this standard emphasises the importance of increasing sensitivity in these diagnostic methods [70]

Microwell and membrane enzyme immunoassays were employed to assess the presence of SwIAV in nasal swabs and lungs from pigs experimentally infected with H1N1 swIAV. However, both tests did not yield reliable results when compared to classical methods [82].

The ESPLINE^®^ INFLUENZA A&B-N kit demonstrated its efficacy by successfully detecting viral antigens in nasal swabs from miniature pigs infected with swine and avian influenza viruses. This kit was deemed sensitive and specific enough for rapid diagnosis of infections with influenza A virus in both chickens and pigs [83].

Rapid test kits for SwIAV diagnosis hold general informative value when there is suspicion of the disease in a herd, but their results should be confirmed by more reliable and accurate methods such as the HA test and RT-PCR.

Commercially available antigen ELISAs are designed for the swift detection of influenza A-type viruses, primarily targeting the conserved nucleoprotein (NP). The ID Screen^®^ Influenza A Antigen Capture kit is a rapid ELISA test specifically created for detecting avian influenza A virus nucleoprotein in samples from birds, pigs or horses. According to an internal validation report (ID.vet) [84], the specificity study involving 200 samples from disease-free chickens yielded negative results. However, there is a lack of specificity studies on pigs. Results from the sensitivity study indicated that the competitor’s immunochromatographic kit detected up to 0.47 ng of Antigen A, while the ID Screen^®^ Influenza A Antigen Capture ELISA detected up to 0.3 ng. In a subsequent experiment, ten different allantoic fluid samples were analysed, and strains from various origins (human, porcine, and avian) were correctly detected. This kit exhibits high sensitivity, high specificity, and high species specificity for influenza A.

The development of the Enzyme-Linked AptaSorbent Analysis (ELISA) antigen for the detection of the SwIAV virus in field samples was conducted by scientists in Spain. Three different aptamers were employed to create an enzyme-linked aptasorbent assay, combined with specific monoclonal antibodies for the detection of influenza A. The study involved the evaluation of 171 field samples (nasal swabs) and achieved a sensitivity of 79.7% and specificity of 98.1%, with real-time RT-PCR serving as the standard assay [85].

## 4. Discussion

As seen in this review, the approaches to diagnostic assays depend heavily on the specific disease under consideration. Therefore, the diagnosis of ASF or SwIA is approached differently, with varying requirements. Economic factors and the geographical situation of areas where outbreaks of the disease may occur and spread to other animals also play a significant role. For example, stringent quarantine and testing of ASF requires rapid and accurate laboratory diagnosis to provide significant data for epidemiology enabling early detection of the disease and consequently, reducing the spread of the disease. The optimum diagnosis of ASF should have consistently high sensitivity and specificity, easy handling and high-throughput application.

ASFV detection methods recommended by the WOAH are virus isolation, antigen measurement using fluorescent antibody tests (FAT) and viral genome detection via polymerase chain reaction. Virus isolation is considered the gold standard diagnostic technique for ASFV; however, it is time-consuming and labour-intensive [43].

Although virus detection is considered the most sensitive technique for ASFV, ELISA has the advantage of its low cost, with competitive sensitive and specificity parameters, and requires less instrumentation than PCR-based diagnosis. For this reason, it is extensively used for large-scale batch detection on pig farms, so they are recommended as the main method for detecting ASFV antibodies. The three commercial kits are based on p30, p72 or pp62 proteins, which are expressed at different ASFV infection stages. Because each ASFV protein provides different characteristics, it is necessary to continuously investigate different antigen combinations to improve the ELISA. For this reason, some dual-antigen ELISA kits include two viral proteins [86]. Some studies include p30 and p54 proteins as potential indicators [87]. The last studies using new antigens include core–shell protein p15 [88], transmembrane protein pI329L [83] or K205R protein [89]

Nowadays, lateral flow tests (LFTs) based on ASFV antigen or antibody detection can be a first-line diagnosis in emergency situations. The sensitivity of this test is limited, thereby it should be used in combination with confirmatory tests. This rapid and simple detection method can be used for early on-site detection of ASFV infection, in places where laboratory equipment is very simple or lacking. Dual antigen or antibody LFT can be also a good option, such as the Anigen ASFV Ag Rapid Test (Bionote). ASFV on-site detection method based on CRISPR/Cas12a technology and LFT has been also designed to detect clinical samples in the field [90].

The number of ASFV strains has increased in recent years, challenging the detection of classical ASFV-p72-based tests. Thus, research on new candidates for the identification of ASFV infection, such as the *MGF505-7R* gene, becomes more important [91]. Indeed, the use of combined assays for portable detection of ASFV has been developed. These molecular systems include the RPA-CRISPR assay, which combines the cleavage of a specific Cas with isothermal amplification RPA assay and a fluorophore-quencher reporter [92]. To improve this method, a LAMP-CRISPR assay has been probed in clinical samples [93]

Overall, ASF diagnosis should be tailored to the epidemiological status of each country/region. ASF-free countries should prioritise epidemiological campaigns to monitor imported animals. In such cases, the advantage of large-scale rapid tests becomes apparent, prioritising ease of use and efficacy. These tests offer reduced impact on transport and customs practices. On the other hand, serological tests offer a relevant and cost-effective method for monitoring a large number of samples from neighbouring zones to affected areas. In contrast, qPCR can accurately identify clinically suspected cases with both specificity and sensibility.

Additionally, other techniques are developed for detecting not just one infection but multiple virus infections in swine. This is the use of multiplex PCR, with no cross-reactivity [94,95]. This Bio.Plex system is important for the implementation of liquid chip technology in diagnosis although it is currently too expensive for routine use.

On the other hand, the scenario for SwIAVs is very different. Classical methods recommended by WOAH for detection and subtyping SwIAV are time-consuming. Therefore, differential real-time RT-PCR is the fastest and most convenient method. The conjunction of the VetMAX™-Gold SIV Detection kit and VetMAX™-Gold SIV Subtyping kit provides a rapid and efficient diagnosis and identification of SwIAV subtypes [70]. Modern methods of gene sequencing are now widely used in research and reference laboratories to determine the subtype and genotype of influenza A virus in pigs, birds and humans, and to analyse their possible new variants [96,97,98,99,100,101,102,103].

When compared to ASFV, SwIAVs exhibit a different epizoological pattern based on their distribution in various regions of the world, being SwIAVs enzootic and geographically dependent. Since 1979, three subtypes, avian-like H1N1, reassortant H1N2 and H3N2 viruses—have been co-circulating in European swine herds. Before 1998, classical H1N1 viruses were the exclusive cause of swine influenza in North America. However, after that, three triple-reassortant viruses—H1N2, H3N2 and H1N1—with genes of human, swine and avian viruses, began to emerge in pigs [63,104].

The presence of serological cross-reactivity among influenza viruses in pigs can complicate diagnosis. In 2004, extensive studies on cross-protection between H1N2 and other influenza A virus subtypes endemic to European pigs were conducted by Dr. Reeth V. et al. These studies revealed a slight antigenic cross-reactivity between subtypes, and the probability of this cross-reactivity increases with increasing antibody levels [105]. Other scientists have also confirmed the presence of serum cross-reactivity between different subtypes of swine influenza viruses [106,107].

Given the previous information, the use of commercially available ELISA kits specifically designed to detect antibodies against SwIAV nucleoproteins (NPs) is recommended. These kits offer a practical and efficient method for assessing the presence of antibodies against SwIAV. They play a crucial role in serological surveillance and diagnostic procedures for swine influenza, providing valuable information for monitoring and managing the health of swine populations.

SwIAV exhibits a remarkable capacity for genetic reassortment, leading to the emergence of novel strains with varying degrees of virulence and transmissibility. This genetic diversity poses challenges for surveillance, diagnostic methodologies, and vaccine development. Moreover, the interconnectedness of human and swine populations emphasises the importance of a One Health approach. Monitoring swine influenza not only contributes to swine health but also plays a pivotal role in preventing potential spillover events and safeguarding public health. Advancements in diagnostic tools are instrumental in enhancing our ability to detect and characterise swine influenza viruses. However, continuous research efforts and international collaboration are essential to stay ahead of the evolving nature of these viruses.

## 5. Conclusions

In conclusion, a thorough analysis of the available information reveals a striking disparity between the number of commercial kits for ASF diagnosis and those for SwIAV. Recent advancements in rapid tests for ASF, improving sensitivity and specificity, accelerate epizootic measures at the point of origin of the disease. Swine influenza diagnosis is in constant flux due to mutations or gene reassortments. Consequently, there is a continual need for monitoring the genomic sequence of newly isolated strains and the development of new diagnostics.

## Figures and Tables

**Table 1 viruses-16-00505-t001:** Specificity and sensitivity of the commercial kits and Taq mix for the detection of the ASFV genome used in the analysis of Schoder et al.

Name of Taq Mixes or Kits	Manufacturer	Specificity	Sensitivity
AgPath-ID™ One-Step RT-PCR Reagents (Ampli Taq Gold™ DNA polymerase)	Applied Biosystems™	100%	100%
TaqPathTM 1-Step Multiplex Master Mix (Ampli Taq™ DNA polymerase)	Thermo Fisher	100%	100%
SsoAdvanced Universal Probes Supermix (Sso7d fusion polymerase)	Bio-Rad	98.96%	97.20%
Virotype ASFV 2.0 PCR kit	Indical	98.45%	98.13%
Adiavet ASFV Fast Time	Adiagen	100%	99.07%
Bio-T kit ASFV	Biosellal	99.48%	99.07%
VetMax ASFV Detection kit	Thermofisher	98.13%	98.45%
RealPCR ASFV DNA Test	IDEXX	100%	98.96%
VetAlert ASF PCR Test kit	Tetracore	98.96%	97.20%
ID Gene™ African Swine Fever Duplex	ID.vet	99.14%	98.67%

**Table 2 viruses-16-00505-t002:** The specificity of diagnostic ELISA kits for antibody detection of ASF.

ELISA Commercial Kits	Technical Basis of the Kit	Specificity
INgezim PPA Compac K3 (Gold Standard Diagnostics, Madrid, Spain)	Blocking ELISA which uses monoclonal antibody of the p72 ASFV protein	84.3%
ID Screen African swine fever indirect assay (IDvet, Grabels, France)	An indirect multi-antigen ELISA kit for the detection of antibodies against p32, p62, and p72 ASFV proteins	100.0%
Svanovir ASFV-Ab; (Boehringer Ingelheim Svanova, Uppsala, Sweden)	Based on indirect ELISA based on recombinant p30 ASFV protein	91.4%

**Table 3 viruses-16-00505-t003:** Diagnostic sensitivity and specificity of PoC tests for the rapid detection of ASF.

PoC Tests, Manufacturer	References	DSe	DSp
INgezim ASF CROM Ag, Gold Standard Diagnostics	[49]	67.86%	97.98%
INgezim ASFV-CSFV CROM Ab, Gold Standard Diagnostics	[50]	100%/96% for ASFV/CSFV	100% for both ASFV/CSFV.
INgezim PPA CROM, Gold Standard Diagnostics	[51]	81.8%	95.9%
Rapid ASFV Ag, Bionote	[52]	Not available	Not available
Rapid Screening Test for ASFV, PenCheckTM	[53]	-	99.4%
SLB ASF Antigen Detection RDT, SLB Co	[54]	65%	76%
GDX70-2 Herdscreen^®^ ASF Antibody, Global DX	[55]	86.2%	100%

**Table 4 viruses-16-00505-t004:** Diagnostic sensitivity and diagnostic specificity VetMAX-Gold SIV solutions.

Commercial Name of Kits	Manufacturer	Specificity	Sensitivity
VetMAX™-Gold SIV Detection kit	Applied Biosystems™	99.1%	98.4%
VetMAX™-Gold SIV Subtyping kit	Applied Biosystems™	100.0%	98.2%

## Data Availability

The data presented in this study are available in public domain resources.

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
