# Peer review of "Overview of Modern Commercial Kits for Laboratory Diagnosis of African Swine Fever and Swine Influenza A Viruses"

_viruses, 2024, doi:10.3390/v16040505_

Round 1

Reviewer 1 Report

Comments and Suggestions for Authors

Traditionally almost every paper about ASF start with "no effective 17 and safe vaccine is available yet", but at this moment it is not clear why authors are so sure that commercial ASF vaccine using in another country are unsafe and not effective. Please provide the reference for this statement or correct the sentence.

I have the same comment to the lines 92 and 93

"This disease has been known to the world community of scientists for more than a 92 century, but the means of treatment and specific prevention have not yet been developed. 93"

In my opinion the information about the number of ASF genotypes is also out of date.

"Currently, 24 genotypes are known by 89 partial sequence analysis of the C-terminal region of the B646L gene encoding the p72 viral 90 capsid protein [21]."

It is another paper published https://www.mdpi.com/1999-4915/15/11/2246

with the different point of view on this classification, and it will be useful, at least mention it, even if authors disagree with it.

"Conventional PCR is also accepted by the WOAH"

"In accordance with the OIE procedure for registration"

Probably authors need to be consistent in the name of organizations in the text.

"Due to the spread of ASFV in Asia, scientists from 173 Chinese institutions have developed a highly sensitive new assay based"

"Chinese institutions" sounds very general. If you mentioned the names of institutes in Ukraine and Spain, you need to do the same for China.

"Due to the lack of 189 vaccination, the presence of antibodies to ASF always indicates infection." Again, information is out of date. Please correct.

"ELISA as well as 194 alternative tests such as indirect fluorescent antibody technique (IFAT), 195 immunoperoxidase technique (IPT) or immunoblot (IB) are the most widely used methods 196 for the detection of ASFV."

I do not think that ELISA allow to detect the virus, please specify the Ab and Ag detection for each method.

"An ASF antigen detection from Shenzhen Lvshiyuan Biotechnology Co. Ltd (SLB) 255" - it looks like the word is missing.

"The Herdscreen® ASFV Ab Test uses a carefully selected ASFV protein. The 263" - it sounds not scientific based, what does it mean "carefully"?

Next two sentences look controversial

"Rapid test kits for the diagnosis of swine IAV have been developed in various 414 countries, but their number,"

"424 Rapid test kits for SwIAV diagnosis have a general informative value when this 425 disease is suspected in a herd and requires confirmation by more reliable and accurate 426 methods, such as HA test and RT-PCR. "

What is the purpose of Rapid test using: diagnosis of screening?

Do the Rapid test kits recommend for diagnosis in any Standards?

Not clear what discussed in Discussion section.

And Conclusions do not connect with the text of the manuscript.

Previously authors mentioned that

"H1 RRT-PCR demonstrated sensitivity and specificity of 100% and 353 99,1%, respectively, for swabs, and 100% and 100% for tissues [60]"

When using a commercially 373 available ELISA that blocks IAV nucleoproteins (NPs) (IDEXX AI MultiS-Screen Ab Test; 374 IDEXX Laboratories, Inc.) and a threshold S/N ≤ 0.60, the sensitivity and specificity were 375 95.5% and 99.6%, respectively [64].

But in Conclusions "the development of new highly specific and sensitive 529 diagnostic kits is an actual and promising direction of research"

What is the gap in the Swine influenza diagnosis if already developed kits have such high specificity and sensitivity?

Author Response

For simplicity, reviewer´s comments are in bold followed by our response.

Comments from Reviewer 1:

  • Traditionally almost every paper about ASF start with "no effective 17 and safe vaccine is available yet", but at this moment it is not clear why authors are so sure that commercial ASF vaccine using in another country are unsafe and not effective. Please provide the reference for this statement or correct the sentence.

We thank the reviewer for this comment and we have incorporated all information about the commercial ASF vaccines used by the Government of Vietnam. We also mention them in several references within the African swine fever section (line 100).

  • I have the same comment to the lines 92 and 93. "This disease has been known to the world community of scientists for more than a 92 century, but the means of treatment and specific prevention have not yet been developed. 93"

We thank the reviewer for this comment and we have incorporated your suggestion in the text (line 92).

  • In my opinion the information about the number of ASF genotypes is also out of date." Currently, 24 genotypes are known by 89 partial sequence analysis of the C-terminal region of the B646L gene encoding the p72 viral 90 capsid protein [21]." It is another paper published https://www.mdpi.com/1999-4915/15/11/2246 with the different point of view on this classification, and it will be useful, at least mention it, even if authors disagree with it.

We thank the reviewer for this comment and we have added this information in the manuscript (lines 97-99).

  • "Conventional PCR is also accepted by the WOAH". "In accordance with the OIE procedure for registration".

Probably authors need to be consistent in the name of organizations in the text.

We agree with this statement and we have changed OIE by WOAH throughout the text.

  • "Due to the spread of ASFV in Asia, scientists from 173 Chinese institutions have developed a highly sensitive new assay based".

"Chinese institutions" sounds very general. If you mentioned the names of institutes in Ukraine and Spain, you need to do the same for China.

Following the reviewer’s comment, we have included this information in the manuscript (line 190).

  • "Due to the lack of 189 vaccination, the presence of antibodies to ASF always indicates infection." Again, information is out of date. Please correct.

We agree with this comment and we have incorporated the corrections (line 207).

  • "ELISA as well as 194 alternative tests such as indirect fluorescent antibody technique (IFAT), 195 immunoperoxidase technique (IPT) or immunoblot (IB) are the most widely used methods 196 for the detection of ASFV."

I do not think that ELISA allow to detect the virus, please specify the Ab and Ag detection for each method.

Following the reviewer’s comment, we have edited this sentence (line 215).

  • "An ASF antigen detection from Shenzhen Lvshiyuan Biotechnology Co. Ltd (SLB) 255" - it looks like the word is missing.

Following the reviewer’s comment, we have included the missing word “kit” (line 270).

  • "The Herdscreen® ASFV Ab Test uses a carefully selected ASFV protein. The 263" - it sounds not scientific based, what does it mean "carefully"?

Following the reviewer’s comment, we have edited this statement (line 278).

  • Next two sentences look controversial.

"Rapid test kits for the diagnosis of swine IAV have been developed in various 414 countries, but their number,"

"424 Rapid test kits for SwIAV diagnosis have a general informative value when this 425 disease is suspected in a herd and requires confirmation by more reliable and accurate 426 methods, such as HA test and RT-PCR. "

What is the purpose of Rapid test using: diagnosis of screening?

Rapid test kits designed for preliminary screening of SwIA viral antigen.

Following the reviewer’s comment, we have changed those sentences (line 433).

  • Do the Rapid test kits recommend for diagnosis in any Standards?

Chapter 3.9.7 of the WOAH contains information about the possibility of using a type A antigen-capture ELISAs and membrane immunoassays for detection of SwIAV in lung tissue and nasal swabs.

We agree with the reviewer and we have added the information (line 435).

  • Not clear what discussed in Discussion section.

Following the reviewer’s comment, we have added a paragraph comparing the situation of ASF with SwIV and their impact in the development of diagnostic tools. This section show discussion for diagnostic strategies approaches for both diseases.

  • And Conclusions do not connect with the text of the manuscript.

Similarly, following the reviewer’s comment this paragraph was edit to summarize the diagnostic approaches for both diseases.

  • Previously authors mentioned that

"H1 RRT-PCR demonstrated sensitivity and specificity of 100% and 353 99,1%, respectively, for swabs, and 100% and 100% for tissues [60]"

When using a commercially 373 available ELISA that blocks IAV nucleoproteins (NPs) (IDEXX AI MultiS-Screen Ab Test; 374 IDEXX Laboratories, Inc.) and a threshold S/N ≤ 0.60, the sensitivity and specificity were 375 95.5% and 99.6%, respectively [64].

But in Conclusions "the development of new highly specific and sensitive 529 diagnostic kits is an actual and promising direction of research"

What is the gap in the Swine influenza diagnosis if already developed kits have such high specificity and sensitivity?

Swine influenza diagnosis is in constant flux due to mutations or gene reassortments. Consequently, there is a continual need for monitoring the genomic sequence of newly isolated strains and the development of suitable vaccines and new diagnostics.

Following the reviewer’s comment, we have added this information in the manuscript (line 563).

Reviewer 2 Report

Comments and Suggestions for Authors

The manuscript “Overview of modern commercial Kits for laboratory diagnosis of African swine fever and swine influenza A viruses” by Larysa Muzykina et al. is an updated review about the current commercial kits used for the diagnosis of ASF and SWIAV. I found it very useful, and I have no further comments. I accept the manuscript in the present form.

Author Response

(The authors gave the same response as above.)

Round 2

Reviewer 1 Report

Comments and Suggestions for Authors

I do not have additional questions or comments.